# Malignant Meningiomas: From Diagnostics to Treatment

**DOI:** 10.3390/diagnostics15050538

**Published:** 2025-02-23

**Authors:** Hojka Rowbottom, Tomaž Šmigoc, Janez Ravnik

**Affiliations:** Department of Neurosurgery, University Medical Centre Maribor, 2000 Maribor, Slovenia; hojka.rowbottom@ukc-mb.si (H.R.); tomaz.smigoc@ukc-mb.si (T.Š.)

**Keywords:** malignant meningiomas, PET SSTR ligands, genetics, DNS methylation profiling, adjuvant radiotherapy, Simpson grade

## Abstract

Meningiomas account for approximately 40% of all primary brain tumors, of which 1.5% are classified as grade 3. Whilst meningiomas are discovered on imaging with high-grade meningiomas being associated with certain imaging features, the final diagnosis is based on histopathology in combination with molecular markers. According to the latest World Health Organization (WHO) Classification of Tumors of the Central Nervous System (CNS), grade 3 should be assigned based on criteria for anaplastic meningiomas, which comprise malignant cytomorphology (anaplasia) that resembles carcinoma, high-grade sarcoma or melanoma; elevated mitotic activity; a TERT promoter mutation and/or a homozygous CDKN2A and/or CDKN2B deletion. Surgery remains the mainstay treatment modality for grade 3 meningiomas, followed by radiotherapy. Limited data are available on the effect of stereotactic radiosurgery and systemic therapy for grade 3 meningiomas; however, studies are underway. Despite optimal treatment, the estimated recurrence rate ranges between 50% and 95% with a 5-year survival rate of 66% and a 10-year estimated survival rate of 14% to 24%.

## 1. Introduction

Meningiomas represent approximately 40% of all primary brain tumors, making them the most common primary intracranial lesion in the adult population with an increasing incidence after the age of 65 years [1,2,3]. According to the latest World Health Organization (WHO) Classification of Tumors of the Central Nervous System (CNS) from 2021, a vast majority (80.1%) of meningiomas are grade 1, followed by grade 2 meningiomas, which present 18.3% of cases, and a mere 1.5% are classified as grade 3 meningiomas, also known as malignant or anaplastic meningiomas [2,4,5]. The incidence of higher-grade meningiomas is greater in Black individuals; however, the reasons remain unknown [1]. A 5-year survival rate of malignant meningiomas is 66% and a 10-year estimated survival is only 14% to 24% [6,7,8].

Ionizing radiation, elevated body mass index (BMI), methotrexate treatment and cigarette smoking have been identified as modifiable risk factors for meningioma development [9,10,11,12,13]. A linear dose–response association has been recognized between the radiation dose and risk of meningioma development in those treated before the age of 10, namely children treated for medulloblastoma or leukemia, who have the highest risk [12,13]. Radiation-induced meningiomas tend to be more aggressive than their spontaneous counterparts with a higher propensity of being grade 2 or 3 and can be multiple [10,14]. Additionally, a positive link has been identified between hormone replacement therapy and the diagnosis of meningioma, as well as the accelerated growth of existing meningiomas during pregnancy [15,16]. Type 2 neurofibromatosis (NF2) is recognized as the most common genetic condition associated with meningiomas, with NF2 patients more often developing grade 2 and 3 meningiomas [17].

## 2. Diagnostics of Malignant Meningiomas

### 2.1. Imaging Diagnostics

Around a fifth of all cases of meningiomas represent an incidental finding on imagining undertaken for unrelated symptoms; however, they can also be symptomatic due to their mass effect or seizures [18,19,20]. Often, the first imaging modality performed is a non-contrast computed tomography (CT) on which approximately a quarter of meningiomas demonstrate some degree of calcification, which can be a sign of slower lesion growth, and thus, a lower WHO grade [21]. Meningiomas are intracranial extra-axial masses with a broad-based dural attachment, or they can appear in a sheet-like form known as “en plaque” meningiomas [22]. In CT scans, meningiomas often appear as homogeneous and hyperdense lobular masses that are enhanced following the application of contrast (extra-axial features) [22,23]. Approximately 50% of all cases of meningiomas are located along the skull base, around 40% are found along the convexity, 10% are located along the falx and parasagittal region and a minority can be found within the ventricles or across multiple locations [24,25]. CT allows for a great evaluation of bony changes, which comprises hyperostosis, which can be either reactive or associated with osseous tumor invasion, osteolysis and, in the anterior skull base, enlarged aerated paranasal sinuses [22].

The imaging modality of choice for the diagnosis of meningiomas remains magnetic resonance imaging (MRI) where the tumor is hypo to isointense to the cortex in T1-weighted sequences and iso to hyperintense in T2-weighted sequences (extra-axial feature). Half of them are linked to perilesional edema (extra-axial features), with the majority of meningiomas enhancing with gadolinium contrast and showing a dural tail (dura-based feature), which can represent reactive dural changes or a region that is also invaded by tumor cells [1,26,27,28,29,30]. As observed with CT and/or MRI, the features linked with high-grade meningiomas (Table 1) are indistinct tumor–brain interface, irregular tumor shape and margins, heterogenous enhancement, larger lesion, absence of calcifications and the presence of perilesional edema [22,31,32,33]. When serial imagining data are available, the volumetric growth rate can also be connected with the grade [34,35,36].

Additionally, diffusion-weighted imaging (DWI) and the apparent diffusion coefficient (ADC), which can provide microstructural information, are important in imaging diagnostics of meningiomas. ADC is relatively low in cases of higher-grade meningiomas [37,38]. Malignant meningiomas are hypercellular, more tightly packed and compact, consisting of cells with a high nucleus-to-cytoplasm ratio, larger nuclei and elevated mitotic activity, which can be appreciated as reduced water diffusivity and, therefore, a lower ADC value [38,39,40]. Studies conducted until now have been inconclusive, with some noting no difference in ADC values and ratios between benign and malignant meningioma, whilst others found significant differences with malignant meningiomas having lower ADC values [40,41,42,43,44]. Diffusion tension imaging (DTI), which analyzes the three-dimensional shape of the diffusion in order to provide information about the direction and the magnitude of water diffusion, demonstrates that in cases of malignant meningiomas with a reduced planar anisotropy coefficient and a higher spherical anisotropy coefficient, this may be a consequence of their loss of normal internal architecture [42,45]. MRI can also reveal a cerebrospinal fluid cleft between the tumor and the brain, which is often absent in higher-grade meningiomas that invade the brain [22].

Additionally, CT/MR-angiogram (CTA, MRA) or CT/MR-venogram (CTV, MRV) can be utilized, especially in cases of skull base meningiomas or lesions near dural venous sinuses to determine the involvement of the vascular structures [46]. Within or around meningiomas, flow voids or enhancing vessels are often present, which can be appreciated on MRI and contrast-enhanced CT as approximately 75% of all meningiomas receive their blood supply from dural vessels or they can receive supply from the carotid or vertebrobasilar circulation through pia [23]. The blood supply is often characterized by a central vascular pedicle and smaller radiating vessels [22,23]. Classic angiography, which shows hypervascularity and a prominent tumor blush with delayed washout, is seldom performed; however, when the non-invasive vascular imaging is unsatisfactory or preoperative embolization is planned, it is still indicated [1,22,47]. Preoperative embolization remains controversial as, on one hand, it reduces blood loss intraoperatively and thus shortens the surgery time; however, it can be linked to an increased risk of postoperative venous thromboembolisms. Therefore, the decision for preoperative embolization has to be individualized [46,47]. Preoperative embolization can be useful in meningiomas, where the feeding artery is inaccessible, such as in petroclival meningiomas where embolization occludes the meningeal branches of the ascending pharyngeal artery or the tentorial branches of the internal carotid artery [47].

In MR spectroscopy, meningiomas are associated with elevated choline and alanine levels and reduced N-acetylaspartate (NAA) levels with elevated alanine being relatively specific for meningiomas [48]. The alanine peak in meningiomas is predicted to be due to the partial oxidation of glutamine or it is converted from pyruvate, which is increased due to the inhibition of the enzyme pyruvate kinase by l-alanine. It is believed that the alanine peak is produced by the transamination of pyruvate in hypoxic tissues to prevent further increases in lactate [49].

Perfusion imaging of meningiomas shows high relative cerebral blood flow (rCBF) and relative cerebral blood volume (rCBV); however, regarding the dynamic susceptibility contrast (DSC), gadolinium leakage can baffle the rCBV quantitation, and less than half of meningiomas return to their baseline of signal intensity after a gadolinium bolus with DSC [50]. On the other hand, some have discovered that higher-grade meningiomas have a significantly lower rCBV, which could be due to continuous relative ischemia and hypoxia with increased rCBV in the peritumoral edema due to its invasion and angiogenesis [51,52]. A clear correlation exists between CBV and the vascularity of the meningiomas as well as between CBV and the expression of the vascular endothelial growth factor, which could help to recognize meningiomas most responsive to anti-angiogenic therapy [51,53]. The Ki-67 proliferative index has been positively correlated with rCBV [54]. The arterial spin labeling (ASL) perfusion of meningiomas is also linked to increased rCBF [55,56]. ASL MRI perfusion provides information about a lesion’s regional flow, enabling a distinction between benign and malignant meningiomas [56].

A novel imaging modality for the diagnosis, surgical resection and radiotherapy treatment volume planning, as well as the post-treatment monitoring of meningiomas, is positron emission tomography (PET) using somatostatin receptor (SSTR) ligands, namely Gallium-68-labeled DOTATATE since the majority of meningiomas express SSTR1/2 [57,58,59]. PET represents an imaging modality that provides biochemical and physiological data about tumors [60]. PET utilizing SSTR ligands has advanced sensitivity compared to MRI regarding meningioma detection, especially in areas of bone invasion (98.5% vs. 52.7%) and regions obscured by calcifications or other radiographic abnormalities, when lesions are located at the skull base or by the falx [61]. Additionally, PET can provide us with an image of the entire body, which allows for the detection of systemic metastases in malignant meningiomas that are rare (estimated prevalence 0.18%), with the lungs, bones and liver being the preferred sites. Often, metastases remain undiagnosed, and with no existing effective treatment plan, prognosis often remains poor [1,62]. PET using Gallium-68-labeled DOTATATE is highly accurate concerning differentiating meningioma from tumor-free tissue, which can be applied to demarcate tumor-invaded bone, which is crucial for surgical planning regarding the extent of drilling as well as allowing for adjustments in radiotherapy planning and assessing the post-radiotherapy response [57,61,63]. The postoperative residual tumor is estimated more accurately using PET compared to MRI since PET can differentiate between a viable tumor and scar tissue using semi-quantitative data analysis [59,64,65]. Fluorine-18-labeled SSTR tracers can also be used for meningioma diagnosis with lower radiation exposure and a longer half-life than Gallium-68-labeled tracers; however, the cost-effectiveness of PET in the diagnosis of meningiomas remains unexplored, as well as the physiological uptake of tracers into nearby anatomic structures and non-neoplastic processes in the rest of the body. Regarding its role in molecular imaging, an evidence-based recommendation in diagnosing meningiomas has been proposed [65,66,67].

In the future, radiomics with the application of large sets of digital medical images and machine learning algorithms could be a great asset in the diagnostics process of malignant meningiomas as well as other intracranial lesions [22,52].

### 2.2. Histopathologic and Molecular Diagnosis with Biomarkers

According to the latest (2021) WHO Classification of CNS Tumors, grade 3 should be assigned based on criteria for anaplastic meningiomas, which comprise malignant cytomorphology (anaplasia) that resembles carcinoma, high-grade sarcoma or melanoma; elevated mitotic activity and a TERT promoter mutation, and/or a homozygous CDKN2A and/or CDKN2B deletion (rhabdoid or papillary histology alone is insufficient) (Table 2). For a meningioma to be classified as having elevated mitotic activity, a mitotic count of ≥12.5 mitoses per mm^2^ (originally noted as ≥20 mitoses per 10 HPF of 0.16 mm^2^) has to be present [4,5,68,69,70,71,72].

The NF2 gene, which remains the most common genetic abnormality in sporadic meningiomas, was the first gene to be linked with the meningioma occurrence and is present in up to 69% of malignant meningiomas [1,73]. NF2, located in chromosome 22q12.2, encodes the protein Merlin, which inhibits signals from the PI3K/Akt, Raf/MEK/ERK and mTOR pathways; thus, the loss of Merlin leads to the overexpression of yes-associated protein 1 (YAP1) and dysregulation of the Hippo-signaling path, leading to an increase in cell proliferation and independent growth with the addition of an increase in apoptotic threshold [74,75,76,77].

In cases of aggressive meningiomas displaying rhabdoid histomorphology, namely rhabdoid cells and a high proliferation index, the inactivation of the tumor suppressor gene (BAP1), which encodes for the breast cancer (BRCA)1-associated protein and can be a somatic loss or a germline mutation, has been linked to poorer prognosis in rhabdoid meningiomas with a reduced time to recurrence. BAP1 germline mutations have been connected with tumor predisposition syndrome (TPDS), predisposing patients mainly to melanoma and mesothelioma [78,79]. In patients with TPDS, malignancies develop in most cases before the age of 55, and most frequently, patients were diagnosed with uveal melanoma, cutaneous melanoma, pleural or peritoneal malignant mesothelioma, or renal cell carcinoma. As a result, in cases of rhabdoid meningiomas, the BAP-1 status has to be assessed [78]. BAP1 loss can be detected by immunohistochemistry and can enable stratification of patient subgroups that require intensive clinical management with close surveillance and consideration for adjuvant therapy [80].

Meningiomas tend to grow in conditions with high levels of progesterone and estrogen, with a large proportion of meningiomas having progesterone receptors, whilst estrogen receptors can be found in less than 10% of cases with the relationship between the WHO staging, prognosis, recurrence and the number of progesterone receptors being inversely related; however, the molecular relation between the occurrence of meningiomas and low levels of progesterone receptors is still unknown and further studies are required to determine the role of sex hormones in the tumorigenesis and progression of meningiomas [81]. Some use mifepristone due to its antiprogesterone activity in the therapy of progesterone receptor-positive meningiomas; however, the results are mixed and inconclusive [82,83].

TERT’s main function is to maintain the DNA telomere ends, and a mutation in TERTp leads to the immortalization of malignant cells [84,85]. The mutation in TERTp can develop during the lesion’s progression and can be limited to an aggressive region of the meningioma; hence, careful tissue sampling for molecular diagnosis is necessary [86,87,88]. Rare mutations, which are associated with more aggressive meningiomas, are ARID1A, PTEN and PBRM1 [73,89,90,91]. Furthermore, the homozygous loss of CDKN2A/B, located in chromosome 9p21, which represents another defining criterion for malignant meningiomas, encodes for several tumor suppressor proteins, such as p16, which prevent the transition of the G1 phase to the S phase in mitosis, leading to the dysregulation of the cell cycle [1,92]. Even the heterozygous deletion of CDKN2A/B is associated with poor outcomes, and meningiomas with increased mRNA expression of CDKN2A/B are also linked to worse prognosis with significant resistance to CDK inhibitors [93].

As well as single-gene alterations, somatic copy number alterations are also connected with meningioma development, namely losses of chromosomal arms 6p, 10q, 14q, 18q, 17q and 20q, which are connected to high-grade meningiomas with losses of 4q, 6 and 19p linked to a worse progression-free survival [94,95,96].

By integrating the contemporary histological grading of meningiomas with prognostic copy number alterations, novel molecular-morphologic grading schemes have developed [1]. The first scheme assigns one point to each of the specific copy number alterations (1p, 3p, 4p/q, 6p/q, 10p/q, 14p/q, 18p/q, 19p/q, CDKN2A/B), which are present with another point for four to nineteen mitoses per 10 high-powered fields or two points for 20 or more mitoses, and meningiomas with four or more points are classified as integrated grade 3 [97]. Another scheme comprises the histologic WHO grade, methylation-class group (benign, intermediate or malignant) and the presence of three prognostic copy number variations (1p, 6q, 14q) [96].

By analyzing the DNA methylation profiles, for meningiomas performed for the first time in 2017, lesions can be distinguished as high and low-risk with six methylation-defined subgroups (benign-1, benign-2, benign-3, intermediate-A, intermediate-B and malignant) of meningiomas, providing better insight into the tumor biology than solely by WHO grade [4,98,99]. Additionally, combining a genome-wide DNA methylation analysis, mRNA expression and copy number alterations, four stable molecular groups (MGs) of meningiomas were invented with MG3 and MG4, including TERTp mutations and homozygous CDKN2A/B deletion, as well as novel somatic mutation in KDM6A, CHD2 and PTEN [100]. MG3 or the hypermetabolic group is linked to the upregulation of several nucleotide and lipid metabolism pathways, whereas MG4 or the proliferative group is associated with changes to the MYC, FOXM1 and E2F pathways, thus leading to disruption of the cell cycle, with MG4 meningiomas harboring the highest mutational and copy number alterations burden [100,101,102,103]. By analyzing the methylation profiles in relation to clinical outcomes and tumor biology, Choudhury et al. developed a classification with three groups of meningiomas, where the hypermitotic (HM) subtype was associated with aggressive behavior and poor outcomes with alterations to the FOXM1 cell proliferation pathways [104]. Further analysis enabled a subdivision of the HM group of meningiomas into a subgroup similar to the MG3 group with changes to the macromolecule metabolism pathways and a subgroup similar to the MG4 group with alterations to cell cycle pathways [105]. Furthermore, Bayley et al. combined DNA methylation analysis, RNA expression, NF2 status and degree of chromosomal instability to develop a system with three methylation groups of meningiomas, where the subtype MenG C meningioma was linked to a higher ratio of copy number alterations and, similarly to the MG3 and MG4 as well as HM groups of meningiomas, was associated with the worst clinical outcomes [106]. Novel molecular classification systems outperform the WHO grading system in predicting clinical outcomes; thus, great efforts are put into expanding access to genomic and methylation testing in all cases of meningiomas [1]. In cases of malignant peripheral nerve sheath tumors, ependymomas and gliomas, as well as other malignancies, the trimethylation of lysine 27 (K27) of histone H3 (H3K27me3) has been found to play an important role in biological processes, such as cell differentiation, proliferation and stem-cell plasticity with histone modifications being recognized in carcinogenesis [98,107,108,109]. In grade 3 meningiomas, however, Nassiri and colleagues found that the loss of H3K27me3 did not affect the outcomes of patients as they found no statistical differences between the groups with retained versus lost H3K27me3 [110]. Furthermore, the Heidelberg study also discovered that a complete loss of H3K27me3 leads to a rapid recurrence in cases of grade 1 and 2 meningiomas, but not grade 3 [111]. The Tübingen study discovered that the loss of H3K27me3, present in 4.7% of 1268 examined meningiomas, was seen in 9.9% of atypical and 16.7% of anaplastic meningiomas, was notably increased in cases of recurrent meningiomas and was also recognized as an independent negative prognostic factor in meningiomas based on a multivariate analysis; however, it was a significant univariate negative prognostic factor for tumor recurrence only in WHO grade 2 meningiomas [112].

Another emerging diagnostic tool for meningiomas is liquid biopsy for the diagnosis and distinction of a meningioma subtype by using cell-free methylated DNA immunoprecipitation and high-throughput sequencing (cfMeDIP) in a plasma sample with extracellular vesicles that had the same methylation characteristics as its parent tumor [113,114]. Plasma-based DNA methylation profiling could also distinguish between cases of benign and malignant meningiomas; however, further validation before daily application is still required [115]. Whilst the detection of elevated protein levels, mutations and epigenetic alterations, as well as changes in RNA and micro RNA (miRNA) expression, can be detected in cerebrospinal fluid, none of the tumor-derived biomarkers, such as circulating tumor DNA, miRNA, proteins and extracellular vesicles, which are used in research, have been validated, and currently, liquid biopsy is not clinically approved for meningiomas [116,117,118,119].

## 3. Management of Malignant Meningiomas

### 3.1. Surgical Management

Surgical resection in grade 3 and 2 meningiomas remains the mainstay treatment modality to this day, with the main goal of resection being a complete neoplasm removal where possible, or a maximal safe resection to reduce the mass effect and alleviate neurologic signs as well as obtaining adequate tissue samples for pathologic diagnosis (Figure 1) [1,70]. To achieve a maximal safe resection that has low morbidity and preserves neurological function, which is important for the outcome, several adjuncts are used during surgery, such as neuronavigation, intraoperative neuromonitoring and intraoperative ultrasound [1]. Additionally, high-definition exoscope systems, which allow for a large focal distance and wide field of view, are also among the notable technological advances making meningioma surgery safer [120]. Prior to surgery, tumor location, consistency, size and proximity to critical structures have to be taken into account in order to plan the correct surgical approach and extent of resection [70]. When a complete resection is not feasible, which can be due to the tumor’s location near neurovascular structures, a maximal possible resection has to be achieved, and the Simpson grade is used to assess the extent of resection [121]. Higher Simpson grades are connected with higher rates of tumor recurrence [121,122,123]. Simpson grades 1, 2 or 3 represent complete tumor resection with resected underlying dura, dura coagulated in situ or dura left intact, respectively. In today’s meningioma surgery, the role of Simpson grading has become controversial, since obtaining grade 1 resection with resection of the underlying dura may not be associated with an improved outcome when compared to other grades [122,123,124,125].

Often, in cases of skull bases meningiomas, a complete resection is unobtainable or linked to increased complication rates, namely cerebrospinal fluid leak, venous infarction or air embolism; hence, a maximal lesion resection with preservation of the underlying dura may lower the morbidity whilst not affecting the progression-free survival (PFS) [122,125]. In cases of convexity meningiomas, a “Simpson grade 0”, in which an additional 2-centimeter margin is being resected, can be achieved [126]. Raman spectroscopy or PET SSTR might = in the future be able to provide us with better guidelines on the extent of dural resection for a prolonged time to recurrence [57,61,127,128]. Given the limitation of the Simpson grading, some have started to use a simpler classification of the extent of meningioma resection with either a gross total resection (GTR), where all of the tumor tissue is removed regardless of handling of the underlying dura (analogous to Simpson grade 1 to 3) or subtotal resection (STR), where a portion of the tumor tissue is left behind (analogous to Simpson grade 4 and 5) [1,70]. This definition of the extent of resection has been adopted by the European Organization for Research and Treatment of Cancer (EORTC) as well as the Radiation Therapy Oncology Group (RTOG) [129]. A 5-year recurrence rate after GTR of grade 3 meningiomas is between 72% and 78% [130,131,132].

### 3.2. Radiotherapy and Radiosurgery

Radiotherapy is currently recognized as the only treatment modality in addition to surgical resection that is recommended for grade 3 meningioma treatment [1]. Adjuvant radiotherapy is indicated in all cases of grade 3 meningiomas, which can be supported by the outcomes of the RTOG-0539 trial with the application of intensity-modulated radiotherapy (IMRT) with 60 Gy over 30 fractions [133]. On the other hand, in cases of grade 2 meningiomas, the evidence of the role of radiation is not clear, and decisions are often made based on physician preference [3].

With novel improvements in the field of radiotherapy, dose escalation in cases of grade 3 meningiomas has been proposed, but further studies are required [1]. A large, single-center retrospective study conducted in Toronto discovered that in cases of grade 3 meningiomas, the dose escalation of conventional photon-based radiotherapy to 66–70 Gy over 33–35 fractions was associated with improvements in local control and the progression-free survival period when compared to standard dose radiotherapy, but with no statistically significant differences in adverse events [134]. At the moment, further randomized controlled trials are needed to determine the most optimal dose of radiotherapy for grade 3 meningiomas [1].

Stereotactic radiosurgery, which is often utilized in small (less than 3 cm) grade 2 meningiomas or their residuals, can also be applied in cases of grade 3 meningiomas; however, evidence is limited with tumor control after stereotactic radiosurgery being relatively poor with rates as low as 17% at 2–2.5 years following radiosurgery. Data about the effects of higher doses or hypofractionated treatment regimens are also poor, and thus, further prospective studies are vital [1,3].

### 3.3. Systemic Therapy

The treatment of grade 3 meningiomas is based on surgery and radiotherapy; however, new systemic agents, namely tyrosine kinase inhibitors and monoclonal antibodies targeting vascular endothelial growth factor (VEGF), have been developed and are under further investigation [135,136,137,138,139]. Sunitinib, a multikinase inhibitor targeting VEGF and platelet-derived growth factor receptors, has been tested in a phase II trial in cases of grade 3 meningiomas with a progression-free survival rate of 42% [136]. Bevacizumab, a monoclonal antibody against VEGF-A, has also been tested in a phase II trial in cases of recurrent grade 3 meningiomas with a progression-free survival rate of 46%, hence suggesting anti-tumor activity [140]. So far, somatostatin agonists and progesterone antagonists and cytotoxic and hormonal agents have not been highly effective in cases of high-grade meningiomas [139,141,142,143]. Another area of possible systemic therapy for meningiomas is immunotherapy and pembrolizumab, a PD-1 inhibitor, which has been trialed for grade 3 meningiomas with a progression-free survival of 48% and a median of 7.6 months; however, 20% of all involved patients developed severe adverse effects [144]. On the other hand, nivolumab failed to demonstrate improvements in progression-free survival [145]. Thus far, no conclusive evidence is available on the application of immunotherapy in cases of grade 3 meningiomas and further trials are required [1]. In cases of NF2-mutant skull base meningiomas refractory to surgery and radiation, currently, there is no effective chemotherapy; however, PIK3CA, SMO and AKT1 represent promising future therapeutic targets [146,147]. Currently, studies using SSTR2 ligands for high-grade recurrent meningiomas are underway with the EORTC Brain Tumor Group network testing ^177^Lu-DOTATATE, as the study NCT03971461 has progressed to the second stage, but further research is crucial at this stage [120,148].

## 4. Outcomes and Prognosis

Cognitive impairments in individuals with grade 3 meningiomas can be present prior to surgery as well as after, especially in domains of memory, attention and executive functions [149,150]. Despite optimal tumor resection and a primarily improved cognitive function after surgery, the long-term health-related quality of life decreases with cognitive and emotional dysfunctions, sleep disorders and fatigue [151,152].

Although surgical treatment and radiotherapy for grade 3 meningiomas have evolved, the estimated 5-year recurrence rate remains high, ranging from 50% to 94% [153]. In cases of recurrence, salvage surgery is the treatment modality of choice, when feasible; however, more than two salvage surgeries are associated with postoperative complications with no benefit for overall survival [154]. According to studies from the United Kingdom and France, the median survival of grade 3 meningiomas is 4 years (ranging between 1.6 and 10.8 years), with a survival rate of 14–34% at 10 years following surgery [8,155]. Male gender and age over 65 years were recognized as predictors of worse overall survival and progression-free survival period in an international multicentric study conducted by Tosefsky et al. [8]. Recurrent grade 3 meningiomas have an even worse prognosis progression-free survival of less than 29% and overall survival of 71% at 6-month follow-up [156].

## 5. Conclusions

Despite recent advances in the field of imaging diagnostics, according to the latest WHO Classification of Tumors of the CNS, histological characteristics and molecular diagnostics remain the basis of diagnosis of grade 3 meningiomas. Up-to-date surgery is considered the mainstay treatment modality for all grade 3 meningiomas receiving radiotherapy, whilst the role of stereotactic radiosurgery and systemic therapy remains elusive.

## Figures and Tables

**Figure 1 diagnostics-15-00538-f001:**
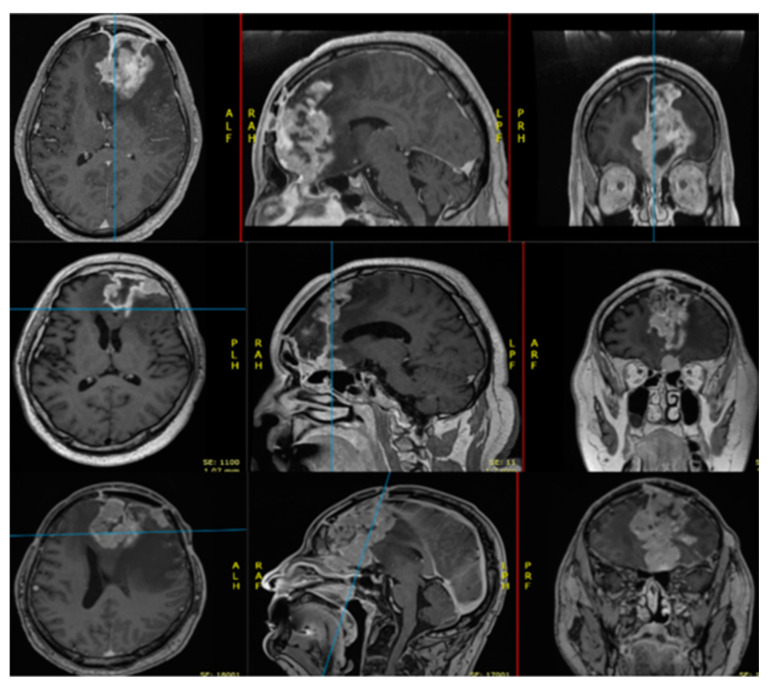
Postoperative MRI (**upper row**) of the brain (axial, sagittal and coronal view after contrast enhancement), performed 6 months after the first resection (Simpson grade 2 resection), showing a 49-year-old male with a diagnosed anaplastic meningioma with a proven loss of CDKN2A and CDKN2B; heterozygous deletion of genes MYCN, PDGFRA and PTEN; amplification of gene PIK3CA and heterozygous deletion of 10q and 19q. The patient was operated on for a second time and the postoperative MRI (**middle row**) of the brain (axial, sagittal and coronal view after contrast enhancement) was performed 2 months after surgery and 4 months after the second surgery (**lower row**), both demonstrating recurrence, and the patient was operated on for the third time. After the first surgery, the patient received irradiation of the tumor bed with a cumulative dose of 66 Gy.

**Table 1 diagnostics-15-00538-t001:** Imaging features associated with high-grade meningiomas.

CT/MRI Features of Grade 3 Meningiomas
Indistinct tumor–brain interface
Irregular tumor shape and margins
Heterogenous enchantment
Larger lesion
Absence of calcifications
Presence of perilesional edema
Volumetric growth rate

**Table 2 diagnostics-15-00538-t002:** Diagnostic criteria for grade 3 meningiomas based on the fifth WHO Classification of Tumors of the CNS.

Diagnostic Criteria for Grade 3 Meningiomas
20 or more mitotic figures in 10 consecutive HPF
*OR*
Frank anaplasia
*OR*
TERT promoter mutation
*OR*
Homozygous deletion of CDKN2A and/or CDKN2B

## Data Availability

The raw data supporting the conclusions of this article will be made available by the authors on request.

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
