# Peer review of "Malignant Meningiomas: From Diagnostics to Treatment"

_diagnostics, 2025, doi:10.3390/diagnostics15050538_

Round 1
Reviewer 1 Report (Previous Reviewer 1)
Comments and Suggestions for Authors
Thank you for addressing the reviewers' comments and questions.
Author Response
Response to Reviewer 1
Thank you for addressing the reviewers' comments and questions.
Response: Thank you so much for your comment and the authors are pleased that our revised work has been satisfactory.
Reviewer 2 Report (Previous Reviewer 2)
Comments and Suggestions for Authors
Author’s extensive revisions in this overall well written and important article are highly appreciated. I have just a few more suggestions for improvement:
- Line 259: “The Tübingen study discovered that in cases of meningiomas, loss of Tübingen was notably increased in recurrent lesions than in primary meningiomas”.. can be replaced along the following lines in author’s own language: “The Tübingen study discovered that the loss of H3K27me3 seen in 4.7% of 1268 meningiomas examined, was identified in 9.9% atypical and 16.7% anaplastic meningiomas, notably increased in recurrent than in primary meningiomas, and was an independent negative prognostic factor in meningiomas on multivariate analysis, but was a significant univariate negative prognostic factor for tumor recurrence only in WHO grade 2 ”
- In table 2, I am not sure why rhabdoid and papillary are mentioned under grade 3 meningiomas. If these are not included in the 2021 WHO classification, should these be removed completely to avoid confusion? Perhaps table 2 should only be captioned “diagnostic criteria for grade 3 meningiomas based on the 5th WHO Classification of Tumors of the CNS” and only list Diagnostic criteria for grade 3 meningiomas (similar to the table given in WHO classification).
- The sentence in Line 276 is addressing two different issues. I suggest place a period (full stop) after recurrence in line 180. Then start a separate sentence as “BAP1 germline mutations have been connected with tumor predisposition syndrome (TPDS)……. Mesothelioma [78,79].”
- Please review a recent article on Molecular classification and grading of meningioma by Nasrallah and Aldape, include any information relevant to your article, and if so, include new reference 155.
- If authors consider appropriate, it may be worth mentioning in line 366 in author’s own language: While there is no chemotherapy to treat NF2-mutant skull base meningiomas refractory to surgery and radiation, PIK3CA, SMO and AKT1 appear to be promising therapeutic targets (references 155, 156).
- Add references if needed:
- 155: Nasrallah MP, Aldape KD. Molecular classification and grading of meningioma. Journal of Neuro-Oncology. 2023 Jan;161(2):373-81.
- Birzu C, Peyre M, Sahm F. Molecular alterations in meningioma: prognostic and therapeutic perspectives. Current opinion in oncology. 2020 Nov 1;32(6):613-22.
Please be careful.
Author Response
Response to Reviewer 2
Authors’ extensive revisions in this overall well written and important article are highly appreciated. I have just a few more suggestions for improvement:
Line 259: “The Tübingen study discovered that in cases of meningiomas, loss of Tübingen was notably increased in recurrent lesions than in primary meningiomas”.. can be replaced along the following lines in author’s own language: “The Tübingen study discovered that the loss of H3K27me3 seen in 4.7% of 1268 meningiomas examined, was identified in 9.9% atypical and 16.7% anaplastic meningiomas, notably increased in recurrent than in primary meningiomas, and was an independent negative prognostic factor in meningiomas on multivariate analysis, but was a significant univariate negative prognostic factor for tumor recurrence only in WHO grade 2 ”
Response: Thank you for your comment. We made corrections and the sentece now reads:«The Tübingen study discovered that the loss of H3K27me3, present in 4.7% of 1268 ex-amined meningiomas, was seen in 9.9% of atypical and 16.7% of anaplastic meningio-mas; notably increased in cases of recurrent meningiomas, and was recognised as an independent negative prognostic factor in meningiomas based on a multivariate analy-sis, however, a significant univariate negative prognostic factor for tumor recurrence only in WHO grade 2 meningiomas« as seen from line 259 to line 264.
In table 2, I am not sure why rhabdoid and papillary are mentioned under grade 3 meningiomas. If these are not included in the 2021 WHO classification, should these be removed completely to avoid confusion? Perhaps table 2 should only be captioned “diagnostic criteria for grade 3 meningiomas based on the 5th WHO Classification of Tumors of the CNS” and only list Diagnostic criteria for grade 3 meningiomas (similar to the table given in WHO classification).
Response: Thank you for your comment. We corrected Table 2 and removed the upper part of it and now it only lists the diagnostic criteria for grade 3 meningiomas.
The sentence in Line 276 addresses two different issues. I suggest placing a period (full stop) after recurrence in line 180. Then start a separate sentence as “BAP1 germline mutations have been connected with tumor predisposition syndrome (TPDS)……. Mesothelioma [78,79].”
Response: Thank you for your comment. We broke the sentence into two parts by placing a full stop where you suggested, and the authors agree that the text flows better.
Please review a recent article on Molecular classification and grading of meningioma by Nasrallah and Aldape, include any information relevant to your article, and if so, include new reference 155. If authors consider appropriate, it may be worth mentioning in line 366 in author’s own language: While there is no chemotherapy to treat NF2-mutant skull base meningiomas refractory to surgery and radiation, PIK3CA, SMO and AKT1 appear to be promising therapeutic targets (references 155, 156).
Add references if needed:
155: Nasrallah MP, Aldape KD. Molecular classification and grading of meningioma. Journal of Neuro-Oncology. 2023 Jan;161(2):373-81.
Birzu C, Peyre M, Sahm F. Molecular alterations in meningioma: prognostic and therapeutic perspectives. Current opinion in oncology. 2020 Nov 1;32(6):613-22.
Response: The authors reviewed both suggested references and found them relevant to our work, which is why we included both references in the revised manuscript. We also added the recommended sentence written in our own words, which reads: “In cases of NF2-mutant skull base meningiomas refractory to surgery and radiation, currently there is no effective chemotherapy, however, PIK3CA, SMO and AKT1 represent promising future therapeutic targets” and can be found from line 365 to line 368.
This manuscript is a resubmission of an earlier submission. The following is a list of the peer review reports and author responses from that submission.
Round 1
Reviewer 1 Report
Comments and Suggestions for Authors
Dr. Rowbottom and colleagues prepared an excellent and comprehensive review manuscript on malignant meningiomas. The review is comprehensive, covering the basic biology, radiology, pathology, and treatment aspects, and supported by ample reference list. It should definitely provide a great resource to anyone who might want to find a concise information on this subject.
I would like to make a few points with the hopes of further improving this manuscript's content. IN no particular order, please consider the following: - Only a few typographical errors are present and can be corrected quickly. For instance, "imaging" ended up as "imagining" in multiple sentences. In line 19 of the Abstract, the sentence says 665, which probably should be 66%. Please spell-check and edit the manuscript. - In the imaging diagnostics, please more explicitly mention dura-based and extra-axial features of meningiomas. These may be straightforward to those who are well-familiar with the diagnostics, but are important features to know for any others hoping to review the subject. - In the short paragraph in lines 106-108, it may be more informative to explain the reason/mechanism of elevated alanine level for meningiomas (similar to the explanations given earlier for ADC and DTI, which are very helpful). - For information where WHO 2021 Classification is mentioned, it may be useful to cite the actual Meningioma Chapter from the Blue Book although the review paper already mentioned is also adequate, but probably less comprehensive for the reader who might want to investigate further. - In lines 153-156, the sentence reads almost like a diagnosis of CNS WHO Grade 3 meningioma cannot be made without molecular support ((probably due to the word "insufficient"). Please update. - In lines 156-157, while accurate, the current official way of reporting mitotic activity is by using mm2. In other words, 10 HPFs each covering a 0.16 mm2 or 12 or more mitotic figures/1 mm2, since different objectives and microscopes may have different area coverages. - In Table 2, "anaplastic" appears like a type of Grade 3 meningioma along with papillary and rhabdoid types, whereas it is simply another name for Grade 3 meningioma. Please clarify the table. - There is a quite comprehensive discussion of molecular genetic aspects of meningiomas; however, a few additional point will strengthen this discussion: 1. BAP-1 loss in some rhabdoid meningiomas and its relation to rhabdoid tumor predisposition syndrome. 2. Status of mostly progesterone and rarely estrogen receptors and their association with and effect on prognosis and treatment. 3. Loss of nuclear H3 K27me3 expression as a worse prognostic sign. - Although briefly mentioned, a stronger word on post-radiation meningiomas tending to be multiple and higher grade, as are seen after craniospinal radiation during childhood for medulloblastoma and leukemic involvement, will be a good addition. - A comment on the extracranial metastases, their incidence, and prognostic and therapeutic implications will be useful. - In general, the typical jump to more aggressive treatment occurs from Grade 1 to high-grade (Grades 2 or 3) meningiomas. It will be good to in turn explain what differences there might be in treatment approaches between Grade 2 and Grade 3 meningiomas. - Speaking of liquid biopsy using blood/plasma, could the authors comment on any information on testing cerebrospinal fluid? - It appears that some references are cited inconsistently to mention only the initials of the Journal. For instance, JNS on line 369, JCO on line 372. Please review the references for consistency.
Reviewer 2 Report
Comments and Suggestions for Authors
Authors have addressed an uncommon but important topic in this overall well written article. In view of the importance of such a review article, and the complexity of the current literature, I request the authors to be more careful with the language and strictly follow the current WHO classification. Accordingly, I have the following few suggestions for improvement:
1. On page 1, Lines 5-14 of Abstract: The authors should rephrase the initial portion of abstract “Meningiomas account for approximately …, diagnosis is based on histological features and molecular markers.” along the following lines in author’s own language within the word limit of the abstract: “Meningiomas account for approximately 40% of all primary brain tumors, of which about 1.5% are classified as grade 3 meningiomas. Meningiomas are discovered on imaging, and while high-grade meningiomas are associated with certain imaging features (especially on recent modalities), however, the final diagnosis is based on histopathology in conjunction with molecular markers. According to the 2021 World Health Organization (WHO) Classification of Tumor of the Central Nervous System (CNS), CNS WHO grade 3 should be primarily assigned by applying the criteria for anaplastic meningioma, namely overtly malignant cytomorphology (anaplasia) that can (1) resemble carcinoma, high-grade sarcoma, or melanoma; (2) display markedly elevated mitotic activity; (3) harbor a TERT promoter mutation; and/or (4) have a homozygous CDKN2A and/or CDKN2B deletion.”
2. On page 4, lines 151-160, authors should rephrase “Grading of meningiomas is based on … and papillary or rhabdoid histology alone is insufficient for a grade 3 meningioma (Table 2) [4,67]” following WHO classification along the following lines in author’s own language: “According to the 2021 World Health Organization (WHO) Classification of Tumor of the Central Nervous System (CNS), CNS WHO grade 3 should be assigned by applying the criteria for anaplastic meningioma, namely overtly malignant cytomorphology (anaplasia) that can (1) resemble carcinoma, high-grade sarcoma, or melanoma; (2) display markedly elevated mitotic activity; (3) harbor a TERT promoter mutation; and/or (4) have a homozygous CDKN2Aand/or CDKN2B deletion (not on the basis of rhabdoid or papillary histology alone) (Table 2) [4,64–67].” A mitotic count of ≥ 12.5 mitoses/mm2 (equating to ≥ 20 mitoses/10 HPF of 0.16 mm2, as originally described) was used to define markedly elevated mitotic activity [4, 134].
3. On page 2, lines 75-78, authors should rephrase “Malignant meningiomas have a more compact … and therefore, a lower ADC value [36–38]” along the following lines in author’s own language “Malignant meningiomas are histologically hypercellular, more compact and tightly packed, comprising of cells with high nucleus-to-cytoplasm ratio, larger nuclei and increased mitotic activity, which in turn correlate with reduced water diffusivity and therefore a lower ADC value on imaging [36–38].”
4. On page 6, lines 231-233, authors should rephrase “The main goal of surgical resection… and relieving neurologic signs and symptoms [1,66]” along the following lines in author’s own language: “Surgical resection remains the mainstay treatment modality to this day. The main goals of resection are complete tumor removal wherever possible, or maximal safe resection to alleviate the mass effect of the tumor and relieve neurologic signs and symptoms; and obtain adequate tissue sample for pathologic diagnosis [1,66]”
5. In the following locations, the word “imagining” has been mistakenly used. Please correct it to “imaging”.
a. Page 1, line 13
b. Page 2, line 47
c. Page 2, line 71
d. Page 2, line 72
e. Page 2, line 73
f. Page 3, line 81
g. Page 3, line 97
h. Page 3, line 123
i. Page 4, line 145
j. Page 9, line 336
6. Page 5, Line 174, replace “considerate” with “careful”.
7. Please review language and structure of sentences throughout the manuscript with a language expert for refinement.
8. Add reference: 134. Perry A, Scheithauer BW, Stafford SL, Lohse CM, Wollan PC. “Malignancy” in meningiomas: a clinicopathologic study of 116 patients, with grading implications. Cancer: Interdisciplinary International Journal of the American Cancer Society. 1999 May 1;85(9):2046-56.
Comments on the Quality of English LanguagePlease work with a language expert for refinement.